# Quercetin and Its Nano-Scale Delivery Systems in Prostate Cancer Therapy: Paving the Way for Cancer Elimination and Reversing Chemoresistance

**DOI:** 10.3390/cancers13071602

**Published:** 2021-03-31

**Authors:** Yaseen Hussain, Sepideh Mirzaei, Milad Ashrafizadeh, Ali Zarrabi, Kiavash Hushmandi, Haroon Khan, Maria Daglia

**Affiliations:** 1Lab of Control Release and Drug Delivery System, College of Pharmaceutical Sciences, Soochow University, Suzhou 215006, China; pharmycc@gmail.com; 2Department of Biology, Faculty of Science, Islamic Azad University, Science and Research Branch, Tehran 1477893855, Iran; sepidehmirzaei.smv@gmail.com; 3Faculty of Engineering and Natural Sciences, Sabanci University, Orta Mahalle, Üniversite Caddesi No. 27, Orhanlı, Tuzla, Istanbul 34956, Turkey; dvm.milad1994@gmail.com; 4Sabanci University Nanotechnology Research and Application Center (SUNUM), Tuzla, Istanbul 34956, Turkey; alizarrabi@sabanciuniv.edu; 5Department of Food Hygiene and Quality Control, Division of Epidemiology, Faculty of Veterinary Medicine, University of Tehran, Tehran 1417466191, Iran; houshmandi.kia7@ut.ac.ir; 6Department of Pharmacy, Abdul Wali Khan University, Mardan 23200, Pakistan; 7Department of Pharmacy, University of Naples Federico II, Via Domenico Montesano 49, 80131 Naples, Italy; maria.daglia@unina.it; 8International Research Center for Food Nutrition and Safety, Jiangsu University, Zhenjiang 212013, China

**Keywords:** quercetin, chemotherapeutic, chemo resistance, mechanisms, nano scale delivery

## Abstract

**Simple Summary:**

Epidemiological studies have shown a negative correlation between the consumption of quercetin and the incidence of prostate cancer, and have indicated a chemo preventive effect of quercetin on prostate cancer in animal models. The major issues associated with quercetin are its low bioavailability and rapid metabolism, and priority attention needs to be addressed to cope with these. Chemoresistance is another of the main negative features concerning prostate cancer treatments. The current review highlights the chemotherapeutic effect, chemo preventive effect, and chemoresistance elimination potential of quercetin in prostate cancer. Quercetin nano scale delivery has been proven to overcome the issues mentioned, however, further studies are required on its nanoscale delivery to make it a next generation agent for the complete eradication of prostate cancer.

**Abstract:**

Prostate cancer is the second most leading and prevalent malignancy around the world, following lung cancer. Prostate cancer is characterized by the uncontrolled growth of cells in the prostate gland. Prostate cancer morbidity and mortality have grown drastically, and intensive prostate cancer care is unlikely to produce adequate outcomes. The synthetic drugs for the treatment of prostate cancer in clinical practice face several challenges. Quercetin is a natural flavonoid found in fruits and vegetables. Apart from its beneficial effects, its plays a key role as an anti-cancer agent. Quercetin has shown anticancer potential, both alone and in combination. Therefore, the current study was designed to collect information from the literature regarding its therapeutic significance in the treatment of prostate cancer. Studies performed both in vitro and in vivo have confirmed that quercetin effectively prevents prostate cancer through different underlying mechanisms. Promising findings have also been achieved in clinical trials regarding the pharmacokinetics and human applications of quercetin. In the meantime, epidemiological studies have shown a negative correlation between the consumption of quercetin and the incidence of prostate cancer, and have indicated a chemopreventive effect of quercetin on prostate cancer in animal models. The major issues associated with quercetin are its low bioavailability and rapid metabolism, and these require priority attention. Chemoresistance is another main negative feature concerning prostate cancer treatment. This review highlights the chemotherapeutic effect, chemo preventive effect, and chemoresistance elimination potential of quercetin in prostate cancer. The underlying mechanisms for elimination of prostate cancer and eradication of resistance, either alone or in combination with other agents, are also discussed. In addition, the nanoscale delivery of quercetin is underpinned along with possible directions for future study.

## 1. Introduction

Throughout the evolutionary history of the human population, awareness regarding plants to be used as herbal medicine was passed on from generation to generation, starting with learning how to pick sources of food as well as how to alleviate illnesses and disease. Allopathic therapies are more common today compared to traditional medicinal products, particularly in developing countries. However, due to the high expense of synthetic drugs, developing countries adopt the use of natural medicines [1,2,3]. The World Health Organization (WHO) states that in developing countries, approximately 80% of the population relies on traditional medicine to fulfill their desired health needs [4].

Elucidating the chemical composition of medicinal plants and their common uses has become a subject of study for many scientific researchers. This research will lead to more new drugs, with minimal side effects compared to existing therapeutic agents [5]. Researchers have also been fascinated by the vast diversity in structures across natural products, and their resultant physicochemical and biological properties. Unfortunately, few plants have been tested for their therapeutic values, and thus the information pool is unable to fully describe the entire potential [6].

The literature on phytochemicals reveals that most active biological constituents from plant extracts are either hydrophobic or hydrophilic, with low membrane penetration capability. This scenario explains the poor bioavailability of these constituents. In addition, the interaction of these phytonutrients within the formulation also presents a challenge. Therefore, maximizing solubility, avoiding degradation of sensitive extracts, and improving absorption (bioavailability) are the main focus required for the use of existing and newly discovered phytochemicals [7,8]. Quercetin is one such phytochemical.

Polyphenol quercetin is an abundant natural flavonoid found in medicinal plants, leaves, vegetables, and fruits. The daily recommended intake of quercetin is 15 mg and it is also recommended as a dietary supplement [9,10]. Quercetin belongs to the flavone class and its structure is characterized by the 5-OH group at the 3, 3′, 4′, 5, and 7 positions (Figure 1) [11]. Quercetin is soluble in organic solvents, but insoluble in water. This hydrophobicity makes it unsuitable for the pharma industry. Many highly advanced techniques have been attempted to crack this barrier including microemulsions, polymeric nanoparticles, solid lipid nanoparticles, liquid crystal systems, liposomes, and precursor systems for quercetin [12,13].

These technologies enable compounds with distinct features to be used within the same formulation, and could even alter the characteristics and activity of a material in biological systems. These advanced drug delivery programs have the capability to not only improve the efficiency of bioactive constituents, but also to reinstitute the ingredients that have been scrapped due to not being effective throughout their formulation [14,15]. Applying nanotechnology to extracts from plants has been extensively described in the literature, as nanostructures could induce the significant activity of plant extracts, facilitate sustained release of bioactive compounds, reduce the required amounts (doses), minimize side effects, and enhance therapeutic potential in the form of enhanced bioavailability [16,17].

Thus, using nanotechnology-based delivery systems is an interesting strategy for optimizing the most desirable properties of a formulation. In addition, nanostructured particles may reflect a future in which the therapeutic potential is assured, and issues pertaining to the use of medicinal plants will be solved.

## 2. Prostate Cancer: Epidemiology, Pathogenesis and Treatment Strategies

The prostate is a male-specific hormone-responsive gland anatomically positioned in the retroperitoneal space and connected physically with the urethra and neck of the bladder. The anatomy of the prostate represents four bio defined zones: (a) the peripheral zone, (b) the central zone, (c) the transitional zone, and (d) the periurethral zone. The prostate is histologically a glandular tissue with a basal layer of low cuboidal epithelial cells protected by a sheet of columnar secretory cells, with ample fibromuscular stroma distinguishing individual glands. Androgens control the growth and survival of cells that make up the prostatic tissue, and eliminating androgens (through castration) contributes to prostate atrophy. In particular, different prostate pathologies appear to occur in different anatomical regions of the gland zones (i.e., the majority of proliferative hyperplastic lesions occur in the transitional zone), and most prostate cancers occur in the peripheral zone. Prostatitis may develop in the absence of persistent infections, termed as chronic a-bacterial prostatitis, or secondary to bacterial infections, known as acute or chronic infection and often presents as a granulomatous lesion [18].

Prostate cancer is the second most prevalent malignancy worldwide, following lung cancer. According to Globocan 2018, the incidence of new cases was 18,078,957 with a total of 9,555,027 deaths in both sexes of all ages, with developed countries showing higher prevalence [19].

Incidence and mortality rates of prostate cancer are closely linked to age, peaking in older people (≥65 years). Note that prevalence rates are significantly higher among African-American men relative to white Americans, with 158.3 reported cases per 100,000 men. and their mortality rate is nearly double that of White men [20]. In 2018, Central America reported the highest mortality rates (10.7 per 100,000 people), preceded by Australia and New Zealand (10.2) and Western Europe (10.1). The minimum rates were recorded in Asian countries (South-Central, 3.3; East, 4.7; and South-East, 5.4) and North Africa (5.8). In addition, Asia held 1/3 (33%) of deaths, followed by 30% in Europe. Prostate cancer mortality increases exponentially with age, and approximately 55 percent of all deaths occurred over the age of 65 [19]. Since 2,293,818 new cases are expected by 2040, a slight increase in death rates is expected (a raise of 1.05 percent) [21].

Differences in social, genetic, and environmental factors have been hypothesized for explanations of this disparity. Numerous studies have confirmed that heritable gene makeup is linked to prostate cancer risk, contributing to complication stats by around 5% [22]. The enzyme ribonuclease is encoded by the HPC1-gene, which is responsible for interferon mediated signaling and defense of the innate immune system [23,24]. Studies of human prostate cancer specimens from ribonuclease gene mutations have revealed the existence of retroviruses that indicate a role for antiviral defenses in the progression of prostate cancer [25]. Thus, it leads toward apoptosis and coping through antiviral activity.

Apart from this, it is suspected that the X chromosome has a role in the inheritance of prostate cancer as it includes the androgen receptor and because minor deletions have been found in the Xq26.3-q27.3 zone in recurrent and inherited types of prostate cancer [26,27]. Clinical and experimental data from a review indicate a link between insulin and prostate cancer, briefly explaining the mechanisms by which insulin is connected to the pathogenesis of prostate cancer. Insulin resistance leads to hyperinsulinemia that in turn results in sympatho-excitatory effects. These effects include alteration in the metabolism of sex hormones, induction of hyperlipidemia, inflammation, and signal transduction that activates insulin like growth factor (IGF) pathways, thus playing a role in the pathogenesis of prostate cancer [28]. Cell proliferation, damage to the DNA, and proteins lead to various effects that further result in prostate cancer (Figure 2).

Over the previous century, treatment methods have been developed for patients with advanced prostate cancer including stage IV hormone-sensitive prostate cancer, recurrent prostate cancer following curative treatment, and castration-resistant prostate cancer, developed extensively with the advent and approval of a range of new drugs including abiraterone, radium-223, enzalutamide, sipuleucel-T, and cabazitaxel, all of which show substantial improvements in overall survival. The proper use of these agents and their proper processing are yet to be established. The findings of several recently published randomized clinical trials and retrospective studies may aid in the development of a treatment strategy for advanced prostate cancer. Furthermore, prospective research on the molecular analysis of tumors are underway to solve these problems [29,30].

Immunotherapy has emerged as a promising treatment choice for prostate cancer patients. Therapeutic cancer vaccines are a potential strategy to cope with the issue. Therapeutic cancer vaccines are engineered to activate immune cells to attack certain tumor-associated antigens that are over-expressed on the surface of cancer cells. These are not specifically associated with substantial toxicity. Some such vaccines are produced via ex-vivo processing that require cell processing, which can be expensive, but can also lead to efficient immune activation. While the vaccines generated are vector based, delivering an immune stimulatory message via subcutaneous injection to the immune system to enable in vivo cytotoxic T lymphocytes. In contrast, a less immune specific strategy may be used, involving injecting irradiated tumor cells into the desired subject, inducing an immune response against the tumor-associated antigens on the surface of the tumor cells [31,32,33]. Despite this promising technique, recent randomized clinical trials using this method have failed to achieve positive clinical output.

Apart from vaccine treatments to mediate the immune response, immune checkpoint inhibitors follow the auto regulatory mechanism, which interferes with the immune system, eventually increasing T-cell function and potentiating antitumor efficacy [34]. A confirmation of the importance of these immune regulatory mechanisms can be observed in the missing immune regulatory mechanisms of CTLA-4 (cytotoxic T-lymphocyte-associated protein 4) knockout mice. The mice died of cytotoxic T lymphocyte (CTL) infiltration of organs within two weeks [35]. A larger scheme of immune-related adverse events is correlated with proven strategies that target the CTLA-4 inhibitory molecule (i.e., panhypophysitis, colitis, and dermatitis), accompanied by collateral damage via CTL activation. Emerging immune checkpoint strategies highlight other molecules, and early findings indicate an immune response with fewer important adverse events [36,37].

Indirect inhibition of tumors through activating the immune system is only one of the variations between immune-based therapies and conventional treatments. Therapeutic cancer vaccines often have limited toxicity, most often injection-site reactions or intermittent fever or flu-like symptoms, associated with the medication. Immune therapies are appealing in non-metastatic and recurrent prostate cancer [38]. Up until now, data on the prevalence and results of prostate cancer, together with a greater understanding of the genetic causes and correlative comorbidities, are important in the prevention and treatment of this cancer.

## 3. Chemoprevention

Natural products and spices have been used since ancient times for the treatment of many illnesses including cancer. The word chemoprevention was invented in the early 1970s and applied to the limited use of phytochemicals for cancer prevention [39]. The idea of using naturally extracted chemicals as potential chemo preventive agents, is a field which has developed significantly over the years, testing a wide variety of agents from natural products using a variety of laboratory models [40]. In addition to a greater understanding of cancer stem cells, emerging developments such as nanotechnology are sure to continue improving the area of cancer chemoprevention in coming years.

Quercetin can directly retard cancer cell growth, and serves as a potent chemo preventive agent for cancer. Quercetin has many intracellular targets within cancerous cells. Many pathways have been postulated to clarify its chemo preventive activity [41]. In various model systems, the chemo preventive effects evoked by this natural molecule are thought to include antioxidant activity, redox homeostasis regulation, cell death, cell cycle arrest, anti-inflammatory action, drug metabolizing enzyme modulation, gene expression pattern changes, Ras gene expression suppression, and signal transduction pathway modulation [42]. Cell signaling networks have gained more attention recently.

Quercetin can affect tumor growth by controlling epigenetics, which can specifically control miRNA expression and DNA methylation levels to exert an anti-cancer effect and increase tumor cell sensitivity to chemotherapy [43]. Quercetin is an ideal therapeutic option to avoid cancer chemotherapy and can directly induce apoptosis in cancer cells without inhibiting normal cell growth. Quercetin, however, has comparatively poor bioavailability, hampering its use as a medicinal agent. If it is eaten as a whole food component, or administered through the use of nanotechnology, the bioavailability of quercetin can be greatly improved for optimal therapeutic outcomes [44].

## 4. Role of Natural Products in Prostate Cancer Therapy

Literature data have shown that certain natural products can specifically target various molecules and signaling pathways involved in the development and progression of tumors. Most of these have been evaluated in both in vitro and in vivo trials, although for some, these clinical trials are currently underway. The preclinical and in vitro studies confirm the outstanding potential of natural products like quercetin, xanthoumol, oridonin fisetin, apigenin, curcumin, resveratrol, genistein, silibinin, kaempferol, honokiol, epigallocatechin-3-gallate, tocotrienols, sulforaphane, luteolin, ginsenosides, ursolic acid, and berberine against prostate cancer [45,46,47]. Atraric acid extracted from Pygeum africanum, acts as an androgen receptor antagonist. It exhibits a potent antiandrogen receptor and anti-prostate cancer activity [48]. Elysia rubefescens is a mollusk containing depsipeptides. An in vitro study has shown that it is effective in solid tumors of prostate cancer cell lines [49]. Hydroxybenzylidene-hydantoin extracted from red sea sponge, Hemimycale arabica, exhibits antimetastatic effects. It has the potential to treat metastatic prostate cancer [50]. Marine bacteria produce certain anti-inflammatory agents like topsentins, manoalide, and pseudopterosins that show anticancer activity in prostate cancer cell lines [51]. 1386A is a marine fungal metabolite. Its activity has been evaluated in prostate cancer cell lines. It manages prostate cancer via inhibition of cancer cell proliferation [52]. Rhizochalin is a marine sponge isolated from Rhizochalina incrustata. Via inhibition of autophagy and induction of apoptosis it can deal with castration resistant prostate cancer [53]. Fucoxanthin is a marine diatom that exhibits potential anti prostate cancer activity [54]. Anthopleura anjunae is a pentapeptide isolated from sea anemone. Its activity was investigated in a DU-145 prostate cancer cell line, where it showed inhibitory effects through interference in the mitochondria mediated apoptotic pathway [55]. 2-((Glycine methyl ester)methyl)-naphtho is a tanshinone analog. It arrests the cell cycle during the G2 phase, inducing apoptosis and metastasis during prostate cancer [56]. Signaling has been found to be biologically active and effective in prostate cancer via the vitamin D receptor. Vitamin D signaling abnormalities can be a target for the treatment and prevention of prostate cancer. Calcitriol deals with prostate cancer following the mechanism of apoptosis, cell cycle arrest, inhibition of proliferative signaling molecules, and modulation of growth factor expression associated with tumors in prostate cancer [57].

Recent literature studies on natural products relating to prostate cancer also reveal that androgen receptors are the main driving forces for the progression and growth of prostate cancer. Thus, modulation of the androgen receptor axis via natural products contributes to prostate cancer therapy. The X chromosome of androgen receptors at the Xq 11-12 loci bears AR genes, which are further composed of N and C-terminal regulatory, DNA, and ligand binding domains. In the absence of androgens, these receptors bind with certain proteins in cytoplasm such as chaperone and heat shock proteins. When ligand binding occurs, they are shifted to the nucleus and undergo homodimerization. Finally, various coregulator proteins and epigenetic factors are recruited which upregulate the downstream gene expression [58,59,60].

Among natural products, flavanols exhibit antitumor activities in addition to their neuroprotective action [61]. Fisetin is a flavanol, which, in the case of prostate cancer, binds to the ligand binding domain, specifically minimizing the stability of androgen receptors and interactions with the carboxyl-amino- terminal. As a result, this decreases the transactivation of androgen receptor genes. In addition to these effects, fisetin also decreases the promotor activity of androgen receptor levels and causes their degradation, which leads to their downregulation. Subsequently, this natural nutrient causes suppression of androgen receptors [62]. Luteolin is a flavone which, in addition to its neuroprotective and anti-inflammatory potential, plays a key role in cancer therapy. In prostate cancer cells, it reduces the protein and mRNA expression of androgen receptors in a time and dose dependent manner. During this process, the level of prostate specific antigens both released and intracellular is reduced by luteolin, contributing to its anti-prostate cancer potential. In addition to this, it causes the destruction of androgen receptors via the proteasome-ubiquitin pathway, further demonstrating its anti-prostate cancer activity [63,64].

Polyphenols from curcuma longa such as curcumin, demonstrate anti-cancer activity. In addition to the down expression of androgen receptor factors, curcumin has also shown its effects in xenografts and prostate cancer cell lines via minimizing the production of testosterone. Where low levels of testosterone downregulate acute steroidogenic regulatory proteins and over expression of aldo-keto reductase, which in turn leads to inactivation of androgenic receptors [65,66,67]. Resveratrol has shown its chemopreventive potential via targeting the androgen receptor axis, as shown in various in vitro models of prostate cancer. It downregulates the hypoxia induced factor alpha (HIF-1α), which inhibits the nuclear translocation of β-catenin, resulting in the inhibition of androgen receptor signaling associated with β-catenin [68,69]. In addition to this, resveratrol also mimics the polyubiquitination of proteasomes of androgen receptor splice variant (ARV7) in the 22RV1 cell line, showing its anti-prostate cancer potential [70]. The information on other natural products such as genistein, celastrol, berberine, honokiol, silymarin, and ginsenosides available in the literature suggests that they may mediate in androgen receptor-based therapy for prostate cancer [71,72,73,74,75,76]. The anti-prostate cancer activity of natural products is mechanistically shown in Figure 3.

Apart from action potential at androgen receptors, multiple natural bioactive compounds have also been documented as exercising growth-suppressive and antiproliferative action in prostate cancer cells and xenografts. Cell survival and growth is associated with the activation of tyrosine kinase receptor–epidermal growth factor receptor (EGFR). Prostate cancer is concerned with the overexpression of epidermal growth factor receptor. Generally, this activates multiple cascade signaling pathways after linking its unique ligands such as epidermal growth factor and transforming growth factor-α including PI3K/Akt/mTOR, mitogen-activated protein kinases (MAPK), hedgehog, and NF-kB [77]. Thus, natural products including berberine, quercetin, luteolin, genistein, and resveratrol suppress the activation of intrinsic tyrosine kinase and ligand-based activation in prostate cancer cells through the reduction of EGFR level [78,79,80,81,82]. In addition, the overexpression of other receptors such as caveolin-1 receptor, zinc dependent mammalian histone deacetylase, PG receptor FP and EP2, prostaglandin degrading enzyme, and prostaglandin endoperoxide synthase protein cyclooxygenase-2 leads to the development of prostate cancer [83,84,85,86]. Quercetin should not be confined as a next generation therapeutic to androgen receptors, but rather should be focused on targeting all these receptor sites.

The insulin growth factor axis is a complex signal transduction system involved in various oncogenic processes, in particular, the proliferation, survival, and metabolism of cancer. The activation of the insulin growth factor axis leads to downregulation of signals in various pathways including PI3K/AKT, MAPK, etc. [87]. Silibinin is a flavonoid, isolated from silybum marianum and the flavone luteolin, which downregulates the overexpression of the IGF-1 domain of the IGF-axis, leading to suppression of the growth and inhibition of the progression of tumors in prostate cancer [88,89,90]. Various natural products are effective in the treatment of prostate cancer, acting on various apoptotic pathways. Berberine induces reactive oxygen species (ROS) in prostate cancer patients, which elicits the translocation of p53 into mitochondria, leading to activation of mitochondrial permeability transition pores. Opening of the mPTP results in the induction of necrosis [91]. Autophagy is a biologically evolved catabolism that the cell typically uses to remove cytosolic components, particularly misfolded proteins and degraded organelles, via a lysosomal degradation process. It includes the formation of double-membrane vesicles known as autophagosomes, which facilitate the recycling of cytoplasmic cargo after lysosomal fusion. In this context, natural products like oridonin, ursolic acid, silibinin, and curcumin trigger cytotoxic autophagy [92,93,94,95].

Progression and growth of prostate cancer is mostly accompanied by metastasis. Cancerous cells experience substantial changes during growth, developing a number of extremely aggressive phenotypes and may become displaced from the original tissue. Epithelial-to-mesenchymal transformation (EMT) is the hallmark of this process, in which a significant shift occurs in the expression of adhesion molecules that control the association of tumor cells with the extracellular matrix and its microenvironment [96,97]. Many natural products have been shown to reverse epithelial-to-mesenchymal transformation in prostate cancer cells and xenografts, particularly by upregulating the signaling cascades PI3K/Akt and Wnt/β-catenin [98,99]. Additionally, epithelial-to-mesenchymal transformation suppression mediated by urokinase-type plasminogen activator, Y-box binding protein-1, and kazal-like proteoglycan-1 domain may be related to the anti-invasive activity of quercetin, fisetin, and apigenin, respectively [100,101,102]. Advanced prostate cancers are accompanied by bone metastasis, causing severe morbidity such as pathological fractures, compression of the spinal cord, and pain. Curcumin was found to suppress bone metastasis in prostate cancer by modulating morphogenic protein-7 as an invasive bone-inhibitor in situ. In addition to curcumin, inhibition of bone metastasis was also seen after treatment with genistein and celastrol [103,104,105].

## 5. Quercetin in Oncology

In recent decades, the scientific community has uncovered the enormous potential role for natural compounds in the therapy and management of terrible diseases such as cancer. Despite the availability of a wide range of natural therapeutic agents, the creation of a definitive treatment for cancer is still pending. Therefore, it is important to understand the relationships between natural molecules and their respective cellular targets to devise an efficient cancer treatment strategy. This would involve numerous intracellular targets, which include apoptosis, cell cycle, detoxification, replication of antioxidants, and angiogenesis. The scope of the synergistic studies available strongly reinforces the use of quercetin as a medication for chemoprevention [106].

Apoptosis is characterized by specific cellular events such as blebbing, failure of cell adhesion, cytoplasmic expansion, fragmentation of DNA, and caspase activation via external and internal pathways. Research indicates that quercetin can induce apoptosis through the mitochondrial pathway involving activation of caspase-3 and 9, accompanied by liberation of cytochrome C and poly-ADP-ribose polymerase cleavage [107,108,109]. This induction of apoptosis by quercetin through mitochondrial pathways and the caspase cascade has been documented in various cancer cell lines including MCF-7 cells of breast cancer, HK1 cells of nasopharyngeal carcinoma, HL60 cells of leukemia, and SCC-9 cells of oral squamous cell carcinoma [110,111,112,113]. Induction of apoptosis via cellular signaling protein modulation, upregulation of Bax (Bcl2 associated X protein), Cox-2, and downregulation of Bcl-2 proteins is also triggered by quercetin [114,115].

Normally, cyclin and cyclin dependent kinases regulate the cell cycle. Conversely, cyclin dependent kinase inhibiter regulates cyclin dependent kinases [116,117]. Quercetin induces S phase cell cycle arrest and subsequently leads to the inhibition of DNA synthesis in SCC-9 cells [111]. During the S-phase of MCF-7 breast cancer cells, quercetin induces cell arrest, which leads to downregulation of cyclin dependent kinases-2 and p53, and p57 upregulation in a dose-time dependent manner [112]. By inhibiting cell cycle progression, quercetin prevents the proliferation of ovarian cancer cells and promotes cell apoptosis [118]. Quercetin anticancer potential evaluated in multiple cancers is briefly shown in Table 1.

Evidence indicates that quercetin deregulates multiple CYP enzyme isoforms in tumor cells [120]. In vitro studies have demonstrated quercetin induced inhibition of CYP1A2 activity in human lung carcinoma A549 cells, HepG2 cells, and human hepatocytes [119]. Apart from CYP enzyme modulation, quercetin follows the mechanism of Nrf2 (nuclear erythroid factor 2-cognate factor 2) mediated enzyme induction, contributing to its anti-cancer potential. When phase II enzymes like heme oxygenase-1, UDP-glucuronosyl, and glutathione S transferases pose any carcinogens, quercetin causes their suppression. The genes of these enzymes involve antioxidant replication components that are rigorously regulated by nuclear erythroid factor 2-cognate factor 2. This, as a consequence, is correlated with another protein known as the Kelch-like ECH-associated protein-1, a Nrf2 repressor, and further reinforces its deterioration via the ubiquitin-dependent proteasome pathway [121,122,123].

In the treatment of ARE-mediated inducer cells, quercetin facilitates the detachment of the Nrf2-Keap1 complex, resulting in the translocation of Nrf2 to the nucleus, where it composes heterodimers with other transcription factors, binds to ARE, and activates phase II enzyme gene transcription [124]. The molecular target pathway is shown in Figure 4.

## 6. Quercetin and Prostate Cancer

Recently, morbidity and mortality of prostate cancer have risen, and systematic cures for prostate are unable to produce sufficient results. Quercetin is a naturally occurring flavonoid compound that has gained immense attention and focus because of its effectiveness against cancer. Both in vitro and in vivo studies have confirmed that quercetin effectively inhibits prostate cancer via different pathways.

### 6.1. Quercetin and Cell Death

Despite the dismal situation in prostate cancer care, the findings of the anticancer effects of quercetin are promising, having been used in a variety of human prostate cancer trials with beneficial effects. During the progression of prostate cancer, quercetin suppresses the epithelial-to-mesenchymal transition process, promoting apoptosis via deactivation of the PI3K/Akt signaling pathway [126]. Additionally, quercetin has been shown to decrease the ratio of Bcl-xL to Bcl-xS and in contrast, maximize the efflux of Bax to the mitochondrial matrix in human prostate cancer cells [115]. Apart from this, quercetin promotes apoptosis of cancer cells by downregulation of heat shock protein-90 levels. Quercetin depletion of heat shock protein-90 results in reduced cell viability, inhibition of surrogate markers, mediated apoptosis, and activation of caspases [109].

A research study on the correlation between quercetin and prostate cancer indicates that quercetin reduces the viability of androgen-independent prostate cancer cells by regulating the expression of system components of insulin-like growth factors (IGF), signal transduction, and inducing apoptosis, which could be very beneficial for the treatment of androgen-independent prostate cancer [127]. There is no study to discuss the role of endoplasmic reticulum stress in quercetin-induced apoptosis in prostate cancer cells. Multiple pieces of evidence indicate several potential signaling pathways for quercetin in apoptosis. In this regard, Liu et al. demonstrated that quercetin decreases the expression of Bcl-2 protein and activates the caspase cascade via mitochondrial and endoplasmic reticulum stress, subsequently leading to apoptosis in prostate cancer cells [128].

Quercetin downregulated the Notch/AKT/mTOR, a fundamental signaling pathway in tumor progression, which leads significantly to apoptosis of U937 leukemia cells [116]. Targeting extrinsic domains, quercetin has been found to boost tumor necrosis factor-related apoptosis-inducing ligand (TRAIL) mediated apoptosis in DU-145 cells (human prostate cancer cell line) via overexpression of death receptor-5 (DR5) [129]. Downregulation of survivin through histone (H-3 regulated) deacetylation and AKT dephosphorylation in prostate cancer-3 and DU-145 cell line also leads to apoptosis by quercetin due to its anti-prostate cancer potential [130,131]. Apart from apoptosis induced by the caspase cascade, quercetin also triggers other apoptosis pathways, which are schematically shown in Figure 5. Apoptosis induction by quercetin, which could be the significant parameter for its anti-prostate cancer effectiveness, has been extensively explored in numerous types of prostate cancer cell and is attracting ever more attention.

### 6.2. Quercetin and Metastasis

The epithelial–mesenchymal transition (EMT) is a flexible transition in the progression of tumors, during which cancer cells undergo drastic changes to develop highly invasive properties. Transforming growth factor-β (TGF-β) is an epithelial–mesenchymal transition inducer within epithelial cells, required for the development of the invasive carcinoma phenotype. Transforming growth factor-β plays a critical role in prostate cancer metastasis and tumorigenesis, with mutations in the Wnt signaling pathway being linked to a further variety of cancer types. Quercetin interferes with the Wnt signaling pathway, leading to inhibition of migration and invasion [132].

Urokinase plasminogen activator (uPA) is a serine protease that is associated with the progression of prostate cancer, especially the invasion and metastasis stages. In the prostate cell proliferation stage, urokinase plasminogen activator is regulated by uPA and transactivation of the epidermal growth factor receptor. Cells of prostate cancer (PC-3) are highly invasive when expressing the uPA and uPAR genes. Quercetin downregulates mRNA expressions for uPA, uPAR, and EGF. In addition, quercetin also inhibits β-catenin, NF-ceB, and even proliferative signaling molecules such as p-EGF-R, N-Ras, Raf-1, c. Fos c. Jun, and p-c. Jun protein expressions of the cell survival factor. This whole process leads to the inhibition of invasion and migration phenomena, resulting in inhibition of prostate cancer metastasis [101]. Quercetin also blocks angiogenesis and metastasis by upregulating thrombospondin-1 to suppress in vitro and in vivo growth of PC-3 cells in human prostate cancer [133].

Angiogenesis is a vital step in the invasion and progression of cancer as it helps the expanding tumor to acquire oxygen and nutrients. At non-toxic concentrations, quercetin significantly inhibits the protrusion of micro vessels and shows substantial inhibition of the proliferation, migration, invasion, and tube forming of endothelial cells, which are essential events in the angiogenesis process. The findings of an associated study revealed that quercetin inhibits angiogenesis and cell growth targeting the VEGF-R2 regulated AKT/mTOR/P70S6K signaling pathway, leading to inhibition of prostate cancer metastasis [134]. Another target for quercetin is miR-21, where it significantly suppresses the proliferation and metastasis of prostate cancer cells and decreases the expression of multiple miRNA associated with prostate tumors, particularly miR-21. Such an inhibition of the miR-21 signaling pathway results in the prevention of prostate cancer metastasis [135]. The comparative detail of quercetin on multiple prostate cancer cell lines, along with the observed effects, are shown in Table 2.

Quercetin in combination with metformin targets the VEGF/PI3k/Akt signaling pathway, which synergistically inhibits cell invasion and proliferation in prostate cancer cell lines [136]. In addition, quercetin in combination with epigallocatechin gallate inhibits the invasion and progression of prostate cancer stem cells via activation of X-linked inhibitor of apoptosis protein (XIAP) and survivin, leading to its metastasis inhibition potential in prostate cancer [137]. With regard to this synergy, in PC-3—the cell lines of human prostate cancer—quercetin and 2-methoxyestradiol display antiproliferative and proapoptotic activity by growing the Stage G2/M of the cell population and decreasing Bcl-2/Bax. Thus, promoting the G2/M stage leads to the anti-metastatic potential of prostate cancer [138]. At low physiological doses, the combination of arctigenin and quercetin targeting related pathways (androgen receptor and PI3K/Akt) offers a novel protocol for accelerated chemoprevention in prostate cancer [139].

### 6.3. Quercetin in Reversing Chemoresistance

Chemotherapy is indeed an indispensable therapy for prostate cancer. The development of chemoresistance, however, is a widespread and crucial issue that requires urgent remedies to be dealt with.

Advanced drug studies have shown that quercetin serves as a potential anti-cancer agent in several types of cancer by regulating multiple pathways. However, current therapies are limited by resistance, which might be reversed by quercetin. In this regard, doxorubicin induced resistance was successfully recovered via quercetin in a research study. A cell line-PC3/R of prostate cancer with acquired doxorubicin resistance was identified. In comparison with normal PC3 cells, a strong drug-resistance to doxorubicin and significant activation of the phosphoinositide 3-kinase/protein kinase-B (PI3K/AKT) pathway was shown in PC3/R cells. Doxorubicin combination therapy with quercetin greatly facilitated the apoptosis induced by doxorubicin in PC3/R cells via the mitochondrial/reaction oxygen species pathway. A major upregulation of tyrosine-protein kinase-met was observed in PC3/R cells as opposed to normal PC3 cells. Furthermore, c-met mediated expression rescued quercetin-promoted apoptosis in doxorubicin treated PC3/R cells [140]. This clearly indicates that quercetin can reverse the resistance of prostate cancer cells to doxorubicin by downregulating the expression of c-met. This might provide a potential strategy to reverse prostate cancer chemoresistance.

Docetaxel is a first line therapeutic drug that is used in the treatment of prostate cancer metastasis. Unfortunately, the advent of resistance reduces its effectiveness and restricts its benefits to survival. In prostate cancer cells and xenograft models, quercetin can reverse docetaxel resistance on proliferation, colony formation, migration, invasion, and apoptosis. Combination therapy of quercetin with docetaxel can sufficiently inhibit the PI3K/Akt pathway and promote apoptosis. Subclones susceptible to docetaxel and prone subclones have been treated with quercetin, which showed that docetaxel-resistant subclones had greater androgen receptor and PI3K/Akt pathway activation, more remarkable phenotypes of mesenchymal and stem-like cells, and more expression of P-gp than that of parental cells. All these transformations were interestingly reversed by quercetin [141]. This offers in-depth evidence for the clinical use of quercetin in docetaxel-resistant prostate cancer.

The effect of cancer treatment and ATP-dependent drug efflux pumps may be significantly affected by multidrug resistance to chemotherapy, P-glycoprotein, and midkine (MK) contribute to the resistance of different chemotherapeutic agents. Z—polypeptide 1 is one of the midkine receptors and, in PI3K and MAPK pathways, has been found to be synergistically active in midkine-mediated cell migration. Consequently, modulation of the PI3K and MAPK signaling pathways by quercetin can cause amplification of gene expression associated with endothelial–mesenchymal transition. Thus, quercetin modulation of the endothelial–mesenchymal transition and drug resistance genes might contribute to the inhibition of CD44^+^/CD133^+^ proliferation and migration [142,143,144]. In summary, these findings show that MK siRNA coupled with quercetin can inhibit the therapeutic resistance of CD44^+^/CD133^+^ cells. Treatment with quercetin combined with the midkine knockdown strategy could effectively target and facilitate removal of CD44^+^/CD133^+^ cells, thereby preventing chemoresistance.

The splice variant AR-V7 is implicated in resistance not only to enzalutamide, but also to abiraterone and other traditional therapeutics. Clinical evidence indicates that resistance toward the next-generation antiandrogen, enzalutamide, can be largely induced by alternative androgen receptor splicing to establish constitutively active splice variants (AR-V7). Recent studies indicate that fusing factors such as hnRNPA1 promote the production of AR-V7 and thus contribute to the resistance of enzalutamide in cells of prostate cancer. Quercetin decreases hnRNPA1, and subsequently AR-V7 expression. Quercetin suppression of AR-V7 desensitizes enzalutamide-resistant prostate cancer cells to enzalutamide therapy. Altogether, the underlying mechanism involves downregulation of hnRNPA1 expression, downregulation of AR-V7 expression, antagonizes the signaling pathway of androgen receptors, and desensitizes enzalutamide-resistant prostate cancer cells to in vivo treatment with enzalutamide in mouse xenografts [145]. These findings indicate that blocking the alternative splicing of the androgen receptor can have major consequences in overwhelming resistance to antiandrogen therapy of the next generation.

Metastatic or locally induced prostate cancer is usually managed with androgen deprivation therapy. Prostate cancer initially reacts to the medication, and then its response begins to revert, gaining tolerance to androgen deprivation and developing toward castrate-resistant prostate cancer-an incurable form. Research using transgenic mouse models shows that modulation of the Wnt/β-Catenin signaling pathway in the prostate cancer is cancerous, allowing for castration-resistant growth of prostate cancer, inducing an epithelial-to-mesenchymal transformation, promoting differentiation of neuroendocrine and giving stem cell-like characteristics to prostate cancer cells [146]. These major Wnt/β-Catenin signaling functions in prostate cancer development emphasize the need to establish drugs targeting this pathway for dealing with resistance to prostate cancer therapy.

## 7. Quercetin and Its Nano Scale Delivery System

In order to solve the problems associated with chemotherapy, nanotechnology-based drug delivery systems have been implemented with beneficial effects on various types of cancers including prostate cancer treatment. Quercetin exhibits certain negative features that lead to its poor systemic availability (Figure 6). Quercetin has poor water solubility (0.00215 g/L at 25 °C to 0.665 g/L at 140 °C) and bioavailability, and it is rapidly metabolized in the body, which can limit its effectiveness as an application for disease prevention or treatment [147,148]. The encapsulation of quercetin into biocompatible and biodegradable nanoparticles may delay or prevent its metabolism, allowing for retention of long-term free levels of quercetin in blood and other tissues. Using nanotechnology-based quercetin formulations can overcome the posed barrier to its delivery.

To cope with the hydrophobicity and poor bioavailability of quercetin in castration resistant prostate cancer, Zhao et al. conducted in vitro and in vivo studies by encapsulating quercetin in nano micelles. An encapsulation of 1 mg/mL efficiently enhanced the water solubility of quercetin 450-fold. The invitro studies showed that the half maximum inhibitory concentration for micellar quercetin formulation was 20 µM, compared to 200 µM of free quercetin. Thus, the nano based formulation efficiently induced apoptosis and inhibited proliferation in human androgen prostate cancer cell lines. In addition, quercetin loaded micelles in vivo displayed superior antitumor efficacy, and proliferation rate decreased by 52.03 percent relative to the control group in the PC-3 xenograft mouse model, likely due to enhanced accumulation of micellar quercetin at the tumor site through enhanced permeability and retention effects [149]. The nano micelle-based drug delivery system forms a promising and successful pharmaceutical treatment approach for prostate cancer (Figure 7).

The serious systemic cytotoxicity of anticancer agents on healthy tissues is one of the main challenges of effective cancer chemotherapy. In this regard, a research study was carried out synthesizing chemically modified poly-lactide-*co*-glycolide (PLGA) nano particles encapsulating a combination of quercetin and docetaxel targeting prostate cancer. The in vitro studies showed higher cellular uptake for quercetin, justified by in vivo activity, which highlights efficient tumor accumulation [150]. Similarly, quercetin triple combination therapy was evaluated to augment the synergistic effect of quercetin. An effective PEGylated niosomes nanostructure was designed encapsulating quercetin, doxorubicin, siRNA, and was targeted in PC-3 cell line of prostate cancer as well as other cancer cell lines. Results of the study show that the aforementioned nano-based delivery system effectively delivers the quercetin cargo into prostate cancer cell lines, leading to quercetin efficient anti-prostate cancer potential [151]. A new avenue seems to have been opened by cationic PEGylated niosomes to expand the therapeutic agent’s armamentarium to combat cancer. The available studies on the nanoscale delivery of quercetin in prostate cancer are shown in Table 3.

Until now, significantly less literature is available on nanotechnology-based delivery of quercetin targeting prostate cancer. However, this needs further research to explore this area. A research work on bladder cancer [152] shows that quercetin nanoparticles remodels the tumor microenvironment and helps in improving synergy. Therefore, in case of combination chemotherapy for prostate cancer, it will be a suitable option to use quercetin along with other potential plant based agents, formulating and targeting it from the nano platform. In addition, prostate cancer is posed to multidrug resistance [153]. To cope with this issue, quercetin is focused on other cancer types [154], but not in prostate cancer from the nanotechnology-based platform. So, its need of the day to evaluate quercetin for multi-drug resistant prostate cancer from a nano-based delivery platform. Apart from this, previous studies on quercetin lipid-based nano delivery shows that it gives efficient release at a pH range of 5–6. Prostate also exhibits a pH in this range, therefore quercetin must be delivered in lipid-based nanoparticles for expecting the aforementioned result.

All of this indicates that targeted delivery of quercetin from the nano platform can be an efficient approach compared to conventional strategies. However, due to limited research work in this area, more attention is needed to explore the anti-prostate cancer activity of this lead compound using nano-technology based approaches.

## 8. Conclusions and Remarks

The massive potential of nanoscale delivery systems is being quickly developed, especially in natural product encapsulation, protection, and delivery. Quercetin is a natural flavonoid that faces various barriers including its poor water solubility, rapid metabolism, and resistance phenomenon. The nanotechnological design and processing of natural products is a significant development path for the global nutraceuticals industry. Quercetin exhibits an immense potential against prostate cancer following various mechanisms such as apoptosis, cell cycle arrest, and inhibition of metastasis as the main underlying ways of cancer prevention. Many therapies including drugs and natural product-based therapy are prone to resistance. Quercetin solves this problem not only via its individual use, but in combination with resisted entities where it aids in overcoming their resistance and ultimately improves therapeutic efficacy. Being hydrophobic, quercetin has lower bioavailability that can be easily overcome by formulating it in nano delivery form. Taking into account current public attention to health problems, the use of natural products for successful disease control and the promotion of health are of paramount importance. In addition, few studies are available on quercetin nano formulations targeting prostate cancer. Researchers must focus on the fabrication of various formulations of nano particles, then investigating their application in prostate cancer, along with the underlying mechanisms. Security and consistency of nanoscale formulations must be of top priority in order to increase market acceptance. Furthermore, future technologies require regulators and manufacturers to tackle safety concerns of quercetin nano-based products prudently through a series of animal and clinical studies to ensure the safety and efficacy of products containing nano delivery systems. This good practice will help mitigate public health and safety issues and increase public awareness of the possible positive health effects of nanoscale quercetin delivery systems targeting prostate cancer.

## Figures and Tables

**Figure 1 cancers-13-01602-f001:**
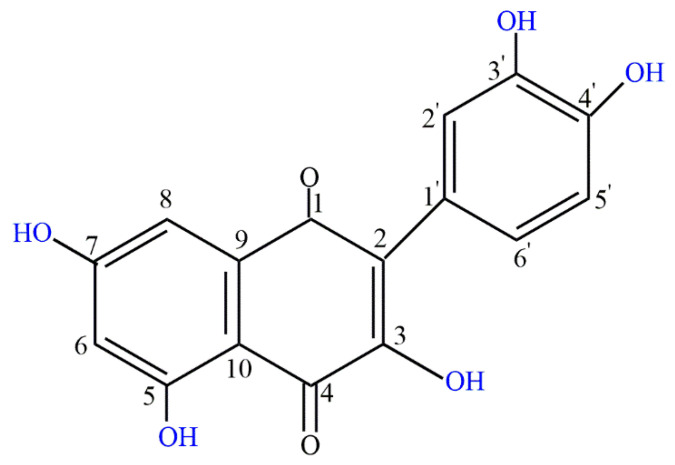
Chemical structure of quercetin.

**Figure 2 cancers-13-01602-f002:**
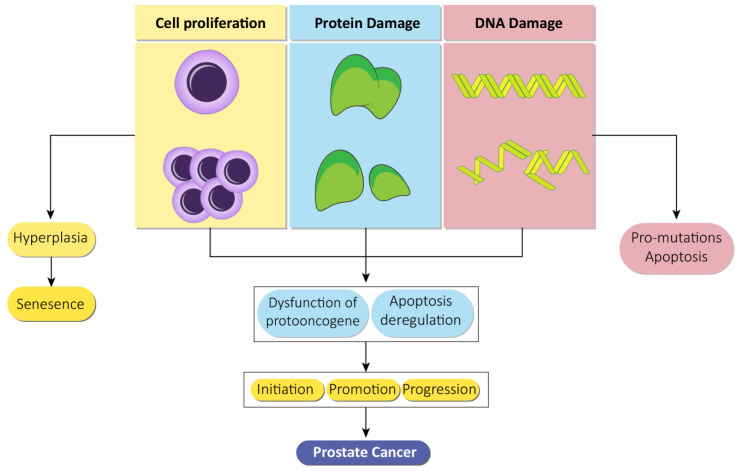
Prostate cancer based on a genotoxic mechanism. Proliferation of cells, damage to proteins, and genetic machinery individually lead to an individual consequence in the form of hyperplasia, cellular senescence, and apoptosis. However, in combination, these lead to multiple effects such as evading repair, protooncogene dysfunction, and apoptosis deregulation that subsequently leads to initiation, promotion, and progression of prostate cancer.

**Figure 3 cancers-13-01602-f003:**
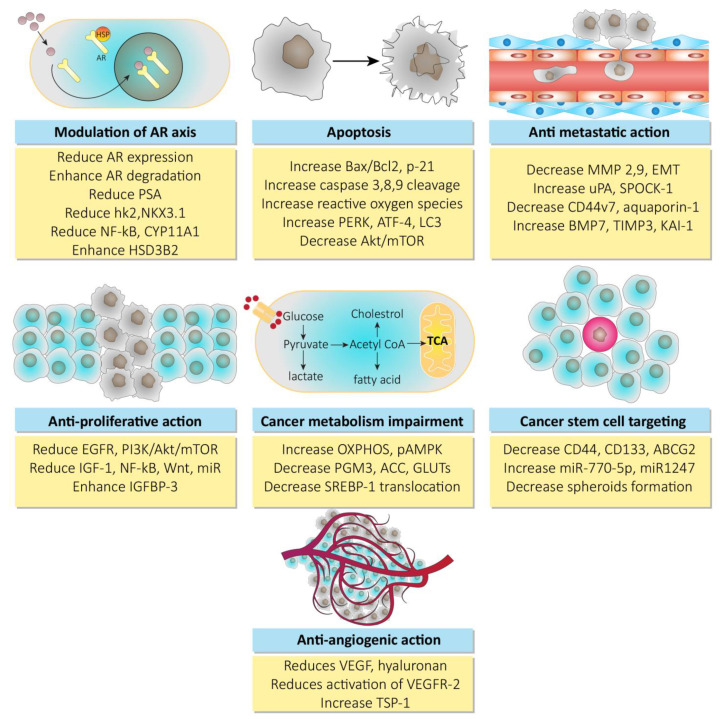
Mechanisms for anti-prostate cancer activity of natural products. Abbreviations: Hsp, heat shock proteins; AR, androgen receptor; PSA, prostate specific antigen; hK2, hexokinase-2; nKX3, homeobox protein; NF-kB, nuclear factor of kappa light chain for B-activated cells; HSD3B2, hydroxy delta-5-steroid dehydrogenase 3-beta delta isomerase 2; EGFR, epidermal growth factor receptor; PI3K, phosphoinositide 3 kinase; Akt, serine/threonine specific protein kinase; mTOR, mammalian target of rapamycin; IGF-1, insulin like growth factor-1; Wnt, wingless int-1; IGFBP3, insulin like growth factor binding protein 3; Bcl-2, B-cell lymphoma-2; PERK, protein kinase RNA like endoplasmic reticulum kinase; ATF-4, activating transcription factor-4; LC3, light chain-3; OXPHOS, oxidative phosphorylation; pAMPK, phosphorylated adenosine monophosphate activated protein kinase; PGM3, phosphoglucomutase 3; ACC, adenoid cystic carcinoma; SREBP-1, sterol regulatory element binding protein-1; MMP, matrix metalloproteinase; EMT, epithelial mesenchymal transition; uPA, urokinase plasminogen activator; SPOCK-1, sparc osteonectin cwcv kazal like domain proteoglycan (testican-1); BMP7, bone morphogenetic protein 7; TIMP3, tissue inhibitor of metalloproteinase 3; ABCG2, ATP binding cassette transporter G-2; VEGF, vascular endothelial growth factor; VEGFR2, vascular endothelial growth factor receptor-2; TSP-1, thrombospondin-1.

**Figure 4 cancers-13-01602-f004:**
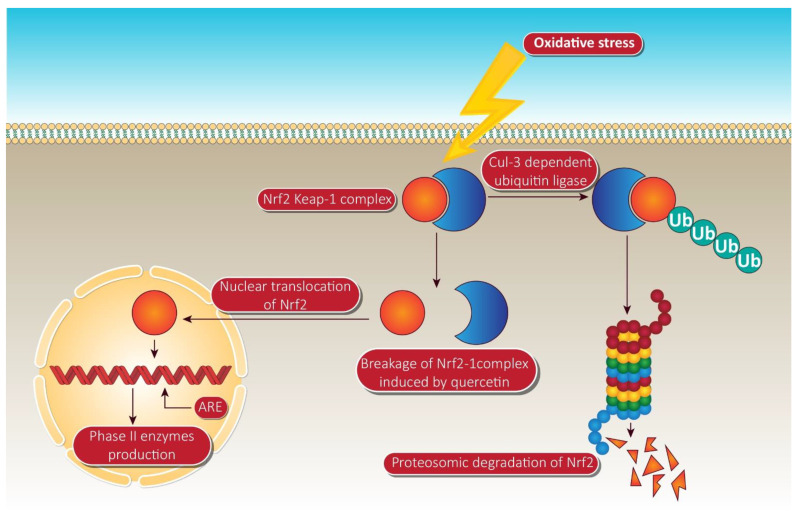
Quercetin targeting via phase II enzyme pathway mediated through Nrf-2. When the cells are unstimulated under stress conditions, Keap-1 isolates Nrf-2 and exposes it to cul-e dependent ubiquitin ligase. This enzyme leads to the proteasomal cleavage of Nrf-2. In the meantime, quercetin degrades the Nrf-2 keap-1 complex and shifts the Nrf2 to the nucleus. Inside the nucleus, Nrf-2 binds to ARE, resulting in the production of phase II enzymes via Nrf-2 associated expression. NrF-2, nuclear factor erythroid 2 related factor-2; Ub, ubiquitin; ARE, anti-oxidant response element. Human colorectal adenocarcinoma cells and duodenal adenocarcinoma HuTu 80 cells were further identified as receiving a quercetin triggered boost in phase II oxidative stress (detoxification enzymes). Additionally, the time-dependent influence of quercetin on the transcriptional regulation of Nrf2 and its increased mRNA and protein expression was consistently observed in HepG2 and malignant mesothelioma cells [125].

**Figure 5 cancers-13-01602-f005:**
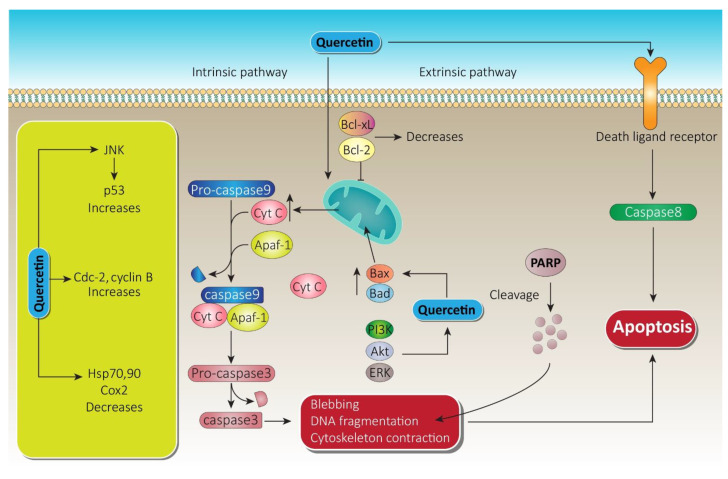
Quercetin apoptotic mechanism via death ligand and mitochondrial membrane. Following the intrinsic pathway, quercetin causes massive release of Cyt-c from the mitochondrial intramembranous space and induces apoptosomes. Furthermore, via blebbing, DNA fragmentation, and cytoskeleton contraction, it paves the way to apoptosis. On the other hand, through the extrinsic pathway, quercetin initiates caspase-8, which leads to apoptosis. A substantial increase in JNK and cdc cyclin B while decreasing heat shock proteins by quercetin also promotes therapy of prostate cancer. Bcl-xL, B-cell lymphoma extra-large; Bcl-2, B-cell lymphoma-2; Cyt-c, cytochrome c; JNK, c-jun N terminal kinase; ERK, extracellular signal regulated kinase; PARP, poly ADP-ribose polymerase; PI3K, phosphoinositide 3 kinase; Akt, serine/threonine specific protein kinase.

**Figure 6 cancers-13-01602-f006:**
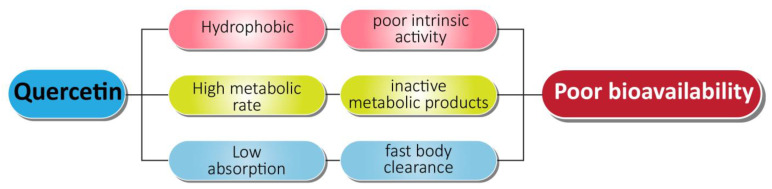
Associated problems with quercetin as an anti-prostate cancer agent.

**Figure 7 cancers-13-01602-f007:**
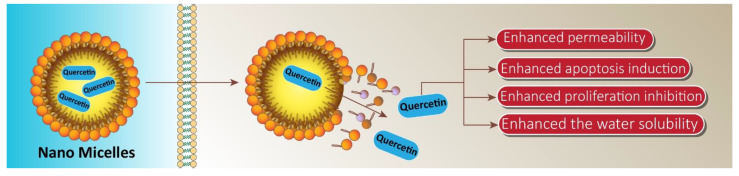
Application of micelles in providing targeted delivery of quercetin in prostate cancer therapy. These nano micelles promote the penetration of quercetin into cancer cells, leading to an increase in bioavailability and subsequent enhancement in suppressing prostate cancer progression and inducing apoptotic cell death.

**Table 1 cancers-13-01602-t001:** Effect of quercetin on multiple cancers.

Cancer Type	Cell Line	Observed Effects	References
Breast cancer	MCF-7 cells	Apoptosis induction, cell cycle arrest	[110,112]
Nasopharyngeal carcinoma	HK-1 cells	Cell cycle arrest and apoptosis induction	[112]
Leukemia	HL-60 cells	Apoptosis induction, detoxification	[113]
oral squamous cell carcinoma	SCC-9 cells	Necrosis and apoptosis induction, cell cycle arrest during S-phase	[111]
Ovarian cancer	SKOV-3 cells	Promotes cell apoptosis, prevents cancer cells proliferation	[118]
Lung cancer	A549 cells	inhibition of CYP1A2 activity	[119]
Gastric cancer	GC 1401	Suppression of gastric cancer cell growth, apoptosis modulation	[1]
Colorectal cancer	HT-229	Apoptosis promotion, provoke cell cycle arrest, proliferation inhibition	[2,3]
Oral cancer	SAS cells	Repression of invasion, migration and cell viability, decrease tumor rate and enhanced apoptosis	[4,5]
Liver cancer	SMMC7721, QGY7701	Antitumor effect via apoptosis induction	[6]
Thyroid cancer	B-CPAP, K1	Promote apoptosis, reduce cell proliferation.	[7,8,9]
Pancreatic cancer	MIA PaCa-2	Apoptosis induction, reduced cell proliferation, apoptosis induction	[10,11]

**Table 2 cancers-13-01602-t002:** Comparative details of quercetin on multiple cell lines of prostate cancer.

Molecular Mechanism	Signaling Pathway	Cell Lines	Observed Effects	References
Apoptosis	PI3K/Akt signaling pathway	PC-3 and its xenograft tumor	Suppression of epithelial to mesenchymal transition	[126]
	Caspase activation, regulation of Bcl-2,	PC-3	Decrease the ratio of Bcl-xL to Bcl-xS and in contrast maximize the efflux of Bax to the mitochondrial matrix	[115]
	Downregulation of heat shock protein-90	PC-3, LNCaP, DU-145	Reduced cell viability, inhibition of surrogate markers, mediated apoptosis and activation of caspases	[109]
	Insulin-like growth factors (IGF), signal transduction both internal and external	PC-3	Reduces the viability of androgen-independent prostate cancer cells	[127]
	Notch/AKT/mTOR, caspase-3, and caspase-9	DU-145	Boost tumor necrosis factor-related apoptosis-inducing ligand (TRAIL), sensitization cancer cells to apoptosis	[129]
Metastasis	Wnt signaling pathway,	PC-3	Inhibition of migration and invasion	[132]
	Inhibition of β-catenin, NF-κB, p-EGF-R, N-Ras, Raf-1, c. Fos c. Jun and p-c. Jun	PC-3	Inhibition of migration and invasion of prostate cancer cell lines	[101]
	Thrombospondin-1	PC-3	Suppress in vitro and in vivo growth of PC-3 cells in human prostate cancer	[133]
Angiogenesis and proliferation	VEGF regulated AKT/mTOR/P70S6K	HUVECs (Human umbilical vein endothelial cells), PC-3	Inhibition of angiogenesis and tumor growth	[134]
	VEGF/PI3k/Akt	LNCap, PC-3	Synergistic inhibition of cell invasion and proliferation	[136]
	capase-3/7, nuclear β-catenin, and TCF-1/LEF	LNCap, PC-3	Inhibition of invasion and proliferation	[137]
	Bcl-2/Bax	LNCap, PC-3	Antiproliferative effect, growing the Stage G2/M	[138]
	PI3K/Akt	LAPC-4 and LNCaP	Inhibition of cell migration, antiprostate cancer potency at lower dose, antiproliferative effect	[139]

**Table 3 cancers-13-01602-t003:** Quercetin and its nanoscale delivery.

Type of Delivery System	Cell Line/Model	Effects/Mechanism	References
Quercetin-loaded nanomicelles	PC-3, human castration-resistant prostate cancer, mouse xenograft model	Proliferation inhibition and apoptosis induction	[149]
Quercetin loaded PEGylated, poly-lactide-co-glycolide nanocapsules	LNCaP/PC-3 cell lines	High caspase-3 activity, improved cell inhibition activity, higher cellular uptake and greater tumor accumulation	[150]
Cationic PEGylated niosome	PC-3 cell line	Higher cytotoxic activity against cancer cells, synergistic effects	[151]

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
