# Peer review of "Quercetin and Its Nano-Scale Delivery Systems in Prostate Cancer Therapy: Paving the Way for Cancer Elimination and Reversing Chemoresistance"

_cancers, 2021, doi:10.3390/cancers13071602_

Round 1

Reviewer 1 Report

Unfortunately, the manuscript does not provide enough scientific results and outcomes and cannot be interesting for the readers of this journal.

1- The subject of the manuscript is very generic.

2- There are many other studies that have been done on this very subject, however, the authors failed to cover those recent advancements in the field.

3- This is more like a summary and not a critical review article.

4- The figures are not well designed and the readers deserve to get the information from high-quality schematic figures.

5- They barely talk about the nanosystems for the delivery of Quercetin. The main scope of this article is nanosystems for delivery of this drug however the review fails to provide the readers with the necessary information.

Regards

Author Response

Reviewer 1

Many thanks for the in-depth analysis of our article and suggesting valuable changes. We have addressed them one-by-one and I am sure that these changes have greatly improved its overall strength of our article. 

Unfortunately, the manuscript does not provide enough scientific results and outcomes and cannot be interesting for the readers of this journal.

  • The subject of the manuscript is very generic.

Author response: we are agreeing with you respected sir but we have tried to compile quercetin and its novel nano-based delivery. In background we have put the generic data.

  • There are many other studies that have been done on this very subject, however, the authors failed to cover those recent advancements in the field.

Author response: thank you for comment. We have put up the latest data from existing literature and also added little recent advancement in the form of novel compounds treating prostate cancer as well as the focus of quercetin on putative receptors other than vitamin D and androgen receptor.

  • This is more like a summary and not a critical review article.

Author response: We are agreeing with you sir, actually we have set the background in such a way that it seems like a full pledge summary. Certain critical points are also discussed in the manuscript.

  • The figures are not well designed and the readers deserve to get the information from high-quality schematic figures.

Author response: As per suggestion, we have made changes to the figures and hope that readers will get interest in it.

  • They barely talk about the nanosystems for the delivery of  The main scope of this article is nanosystems for delivery of this drug however the review fails to provide the readers with the necessary information.

Author response: As per suggestion, all the available data on the nano-based delivery of quercetin is effectively presented in the text. Additionally, further research is emphasized in the conclusion section.

Reviewer 2 Report

The review paper by Dr Yaseen Hussain et al deals with Quercetin and its nano-scale delivery systems in prostate cancer therapy. In particular, the authors discuss the underlying mechanisms for elimination of prostate cancer and eradication of resistance either alone or in combination with other agents

Two minor changes are suggested:

  • The sections 1 (Introduction) and, in particular, 2 (Prostate cancer: Epidemiology, pathogenesis and treatment strategies) could be shortened.
  • A separate section dealing with Chemoprevention should be added.

Author Response

Reviewer 2

Many thanks for the in-depth analysis of our article and suggesting valuable changes. We have addressed them one-by-one and I am sure that these changes have greatly improved its overall strength of our article. 

Two minor changes are suggested:

Reviewer suggestion: The sections 1 (Introduction) and, in particular, 2 (Prostate cancer: Epidemiology, pathogenesis and treatment strategies) could be shortened.

Author response: Thank you for suggestions. We have made changes as per your instructions in both sections.

Reviewer suggestion: A separate section dealing with Chemoprevention should be added.

Author response: We have added the section chemoprevention as per your instructions and the addition is highlighted via yellow color in our manuscript. Your suggested addition will be fruitful for our manuscript.

Reviewer 3 Report

The manuscript by Hussain et al. summarizes in a comprehensive manner the up-to-date knowledge of the natural compound quercetin in prostate cancer treatment. Authors indicate critically the low bioavailability and solubility and indicate the new direction to load compounds on nano-particles. Other recent reviews on quercetin in cancer treatment were recently published but this review focusses on prostate cancer to provide critical insights.

Major points:

It would be important to mention putative Quercetin receptors. Binding of quercetin to some tyrosine kinase receptors and to the vitamin D receptor were reported and could explain the interference of quercetin with the described pathways. Authors may critically include these findings.

Page 5. Authors mention a battery of natural products analyzed in prostate cancer. Meanwhile many novel compounds from herbs, marine and fungi have been isolated that should also be mentioned, including YVPGP-oligopeptide, atraric acid, vitamin D3, etc. Please complete the list.

Authors mention androgen receptors. However, there is primarily one androgen receptor present that drives androgen effects. The splice variants lack the LBD. Thus, use the singular if the full-length receptor is meant.

Minor points:

There are some strange sentences and slang phrases. Authors must edit the manuscript extensively. Only few are listed such as:

Page 1: “occurred from race to race“ replace by:  from generation to generation.

Page 2: “very low level of plants has been tested… “ replace by: only few plants

Page 3: “Thus, contributing toward apoptosis and coping with antiviral activity.“, sentence is not compete.

Fig. 2 legend. Please mention: cellular senescence, which is distinct from organismic aging/senescence.

Page 5: “inhibiters” change to: inhibitors

Page 5: “androgens get binds“ change to: „androgens bind“

Page 6: „Resveratrol acting on androgen receptor axis in various invitro models of prostate cancer showing its chemo preventive potential“ Sentence is strange.

Page 6: “Other natural products like genistein, celastrol, berberine, honokiol, silymarin and ginsenosides are available in literature“ compounds themselves are not available in literature. The information about compounds is available.

Page 9:What is meant by “ARE-mediated inducer cells“?

Page 13: “However, quercetin therapy is limited by resistance that“ I guess it is rather meant that current therapies are limited by resistance which might be reversed by quercetin?

Author Response

Reviewer 3

Many thanks for the in-depth analysis of our article and suggesting valuable changes. We have addressed them one-by-one and I am sure that these changes have greatly improved its overall strength of our article. 

Major points:

Reviewer comment: It would be important to mention putative Quercetin receptors. Binding of quercetin to some tyrosine kinase receptors and to the vitamin D receptor were reported and could explain the interference of quercetin with the described pathways. Authors may critically include these findings.

Author response: Thank you for your informative suggestion and it will improve our manuscript. We have added the desired receptor detail in our manuscript as per your instructions and have been highlighted in the manuscript.

Reviewer comment: Page 5. Authors mention a battery of natural products analyzed in prostate cancer. Meanwhile many novel compounds from herbs, marine and fungi have been isolated that should also be mentioned, including YVPGP-oligopeptide, atraric acid, vitamin D3, etc. Please complete the list.

Author response: we are agreeing with you for informative suggestion. A detail of novel compounds from herbs and marine sources have been added to the manuscript from the available literature and were highlighted.

Reviewer comment: Authors mention androgen receptors. However, there is primarily one androgen receptor present that drives androgen effects. The splice variants lack the LBD. Thus, use the singular if the full-length receptor is meant.

Author response: We have made the changes in this context within the mentioned portion of manuscript.

Minor points:

Reviewer comments: There are some strange sentences and slang phrases. Authors must edit the manuscript extensively. Only few are listed such as:

Page 1: “occurred from race to race“ replace by:  from generation to generation.

Page 2: “very low level of plants has been tested… “ replace by: only few plants

Page 3: “Thus, contributing toward apoptosis and coping with antiviral activity.“, sentence is not compete.

Fig. 2 legend. Please mention: cellular senescence, which is distinct from organismic aging/senescence.

Page 5: “inhibiters” change to: inhibitors

Page 5: “androgens get binds“ change to: „androgens bind“

Author response: Thank you for highlighting minor corrections. We have revised the whole manuscript and corrected the slang and strange sentences as per your instructions. Changes made were highlighted.

Reviewer comment: Page 6: „Resveratrol acting on androgen receptor axis in various invitro models of prostate cancer showing its chemo preventive potential“ Sentence is strange.

Author response: The sentence was strange we are agreeing with you. However, now it has been rephrased and highlighted so will be easily understand by readers.

Reviewer comment Page 6: “Other natural products like genistein, celastrol, berberine, honokiol, silymarin and ginsenosides are available in literature“ compounds themselves are not available in literature. The information about compounds is available.

Author response: we have made changes to the sentence as per your suggestion and changes made were highlighted.

Reviewer comment : Page 9:What is meant by “ARE-mediated inducer cells“?

Author response: ARE-mediated inducer cells mean antioxidant-responsive element, which mediate the transcriptional induction of a battery of genes which comprise much of this chemo protective response system.

Reviewer comment Page 13: “However, quercetin therapy is limited by resistance that “I guess it is rather meant that current therapies are limited by resistance which might be reversed by quercetin?

Author response: we are agreeing with your statement. The sentence was rephrased and highlighted.

Reviewer 4 Report

The review is highly unfocused and authors have taken more room to describe the general issue of prostate cancer which is reviewed by others. Hence not adding any value to this review and not giving any new information to the readers of the Cancers. Captured on Quercetin is not new and reviewed by others. In addition to this manuscript is not well written and the content is not aligned with the title of the review. Hence I feel this manuscript is not suitable for publication in cancers.

Author Response

Reviewer 4

Many thanks for the in-depth analysis of our article and suggesting valuable changes. We have addressed them one-by-one and I am sure that these changes have greatly improved its overall strength of our article. 

Reviewer comment: The review is highly unfocused and authors have taken more room to describe the general issue of prostate cancer which is reviewed by others. Hence not adding any value to this review and not giving any new information to the readers of the Cancers. Captured on Quercetin is not new and reviewed by others. In addition to this manuscript is not well written and the content is not aligned with the title of the review. Hence, I feel this manuscript is not suitable for publication in cancers.

Author response: We have discussed the general issues of prostate cancer for the purpose to set a background for the readers on prostate cancer and then chemoresistance as well. 

Similarly, we have focused the quercetin delivery targeting prostate cancer following nano scale delivery. However, there is less literature available on nano scale delivery of quercetin which we have mentioned in our manuscript for the future.

Addition to this, we have addressed the maximum literature on nano scale delivery of the concern phytochemical. Apart from it, we have made changes to the manuscript especially improved its language, figures etc.

Apart from this, we have revised all the figures and added a new figure to attract readers.

I am sure that these efforts have made the overall article very attractive for the readers and you will like it in the present form.

Reviewer 5 Report

The manuscript by Hussain et al. entitled “Quercetin and its nano-scale delivery systems in prostate cancer therapy: Paving the way for cancer elimination and reversing chemoresistance” is a literature review on the potential therapeutic role of quercetin in prostate cancer. The topic is interesting with a translational impact. However, the review needs to be extensively revised for the following reasons: Although this paper includes a large body of evidence supporting the beneficial role of quercetin, however it is difficult to follow it. Besides, several key terms/concepts should be carefully revised. For instance the term “Dysfunction of protooncogene” is inappropriate. Last the manuscript should be also revised prior re-submission by an English native speaker.

Author Response

Reviewer 5

Many thanks for the in-depth analysis of our article and suggesting valuable changes. We have addressed them one-by-one and I am sure that these changes have greatly improved its overall strength of our article. 

Reviewer comment: the manuscript by Hussain et al. entitled “Quercetin and its nano-scale delivery systems in prostate cancer therapy: Paving the way for cancer elimination and reversing chemoresistance” is a literature review on the potential therapeutic role of quercetin in prostate cancer. The topic is interesting with a translational impact. However, the review needs to be extensively revised for the following reasons: Although this paper includes a large body of evidence supporting the beneficial role of quercetin, however it is difficult to follow it. Besides, several key terms/concepts should be carefully revised. For instance, the term “Dysfunction of protooncogene” is inappropriate. Last the manuscript should be also revised prior re-submission by an English native speaker.

Author response: Thank you very much for the valuable suggestions. We have revised the whole manuscript and changes were made as per your instructions.

All the figures are revised and a new figure is added to attract the readers.

The language and slanged concepts were modified and highlighted in manuscript. The language corrections are made by Prof. Dr. Maria Daglia.   

Round 2

Reviewer 1 Report

To the Authors,

Many thanks for providing the revised version of the manuscript. The current version of manuscript still suffers from many problems. Not having a comparison table in "6. Quercetin and prostate cancer" is one these problem. Unfortunately the body of the manuscript does not completely comply with what the authors claim in the title. The number of works using naosystems for delivery of Quercetin for the treatment of prostate cancer is not sufficient. The major part of this manuscript tries to describe the basics of the drug and prostate cancer, however, the readers are interested in critical points of the treatment of prostate cancer with this drug, and also the outcomes of the clinical trials on this very subject.

Regards 

Author Response

Reviewer 1

Many thanks for providing the revised version of the manuscript. The current version of manuscript still suffers from many problems. Not having a comparison table in "6. Quercetin and prostate cancer" is one these problems. Unfortunately the body of the manuscript does not completely comply with what the authors claim in the title. The number of works using nano systems for delivery of Quercetin for the treatment of prostate cancer is not sufficient. The major part of this manuscript tries to describe the basics of the drug and prostate cancer; however, the readers are interested in critical points of the treatment of prostate cancer with this drug, and also the outcomes of the clinical trials on this very subject.

Author response: A table containing comparative analysis of quercetin-induced effects on prostate cancer types (cancer cell lines) was prepared, which is available on page 14 of manuscript as Table 2.

Addition was made to the nanoscale delivery of quercetin with some future directions that will hopefully take the attention of readers. No sufficient data on quercetin nano-delivery is available in literature. The mentioned data is the most possible collected literature.

Additionally, the nanoformulations of quercetin are not fully explored thus lack clinical trial studies. Changes made were highlighted in green color.

Reviewer 4 Report

This revised review still lacks effective written communication. Written components are not aligned with review title and theme of the review. Authors still fail to highlight the Quercetin for example no description was provided about different from and its metabolites with works. How this agent is chemo preventive as well as Therapeutic. In addition to this there are different delivery system existing why one delivery system is suitable for delivering Quercetin is better over other is missing and not discussed. Further, this review is not adding any thing new or new prospective.

Author Response

Reviewer 4

This revised review still lacks effective written communication. Written components are not aligned with review title and theme of the review. Authors still fail to highlight the Quercetin for example no description was provided about different from and its metabolites with works. How this agent is chemo preventive as well as Therapeutic. In addition to this there are different delivery system existing why one delivery system is suitable for delivering Quercetin is better over other is missing and not discussed. Further, this review is not adding anything new or new prospective.

Response: Many thanks for your opinion/suggestions. However, we have extensively reviewed and modified where required and added new figures for better understanding and attraction of readers. The table is also improved for readers. I am sure the new version will impress you. 

Reviewer 5 Report

The revised manuscript by Hussain et al. entitled “Quercetin and its nano-scale delivery systems in prostate cancer therapy: Paving the way for cancer elimination and reversing chemoresistance” has been substantially improved. The authors could add a Table summarizing the key findings regarding the therapeutic-biological activity of quercetin in different types of cancer.

Author Response

Reviewer 5

The revised manuscript by Hussain et al. entitled “Quercetin and its nano-scale delivery systems in prostate cancer therapy: Paving the way for cancer elimination and reversing chemoresistance” has been substantially improved. The authors could add a Table summarizing the key findings regarding the therapeutic-biological activity of quercetin in different types of cancer.

Author response: the aforementioned table was added as Table 1 on page 10 of the manuscript. The changes made are highlighted in green color.

Round 3

Reviewer 1 Report

Dear authors,

The provided information has elevated the quality of the paper, however, still there are serious flaws in this review article.

  • The article does not provide the readers with the critical data about quercetin.
  • The title of this manuscript does not reflect the main aspects of this work.
  • Extensive editing of English language and style is required. This review article does not critically talks about the nanosystems for delivery of quercetin.
  • The title should not have the nano-scale delivery system in it because there are not sufficient data on this issue.

Regards

Reviewer 4 Report

Authors have given multiple chances to enhance the quality of the manuscript. However, the manuscript still lacks celerity and focus. After several revisions, the focus is still missing, writing is redundant, and most sections are written to incase the content size not to keep the sole of this review. The authors have extended some sections, which is still not a meaningful addition. Major highlights authors stated are chemo-preventive, Therapeutic, and potential of quercetin to revers chemoresistance. However, it’s disappointing to see that authors have failed to revised/added the sections such as chemoprevention and have added additional information in section Role of natural products in prostate cancer therapy to justify the theme authors intended to cover.  However, information added is irrelevant and redundant and is just a statement taken from published work. Even under unnecessary sections such as the immunotherapy sections, authors have not discussed the most recent immune-based therapeutics offered to treat prostate cancer, and without rationalizing if this natural agent has some role impacting the adaptive immune system, I don’t see the logic for the extensive section on this aspect.  Authors have also revised nanoscale delivery sections with limited discussion on the type of delivery, and author has emphasized that a lipid-based delivery system will be better to address bioavailability but has not discussed how a lipid-soluble compound will be delivered at the tumor site and how lipid based delivery system will be able to identify target cells.